# Improving Bias Mitigation through Bias Experts in Natural Language Understanding

**Eojin Jeon**[1*]**, Mingyu Lee**[1*]**, Juhyeong Park**[1]**,**
**Yeachan Kim**[1]**, Wing-Lam Mok**[1]**, SangKeun Lee**[1,2]
[1]Department of Artificial Intelligence [2]Department of Computer Science and Engineering
Korea University, Seoul, Republic of Korea
{skdlcm456, decon9201, johnida, yeachan, wlmokac, yalphy}@korea.ac.kr

## Abstract

Biases in the dataset often enable the model to achieve high performance on in-distribution data, while poorly performing on out-of-distribution data. To mitigate the detrimental effect of the bias on the networks, previous works have proposed debiasing methods that down-weight the biased examples identified by an auxiliary model, which is trained with explicit bias labels. However, finding a type of bias in datasets is a costly process. Therefore, recent studies have attempted to make the auxiliary model biased without the guidance (or annotation) of bias labels, by constraining the model's training environment or the capability of the model itself. Despite the promising debiasing results of recent works, the multi-class learning objective, which has been naively used to train the auxiliary model, may harm the bias mitigation effect due to its regularization effect and competitive nature across classes. As an alternative, we propose a new debiasing framework that introduces binary classifiers between the auxiliary model and the main model, coined bias experts. Specifically, each bias expert is trained on a binary classification task derived from the multi-class classification task via the One-vs-Rest approach. Experimental results demonstrate that our proposed strategy improves the bias identification ability of the auxiliary model. Consequently, our debiased model consistently outperforms the state-of-the-art on various challenge datasets.[1]

## 1 Introduction

Deep neural networks achieve state-of-the-art performances on a variety of multi-class classification tasks, including image classification (He et al., 2016) and natural language inference (Devlin et al., 2019). However, they are often biased towards spurious correlations between inputs and labels, which

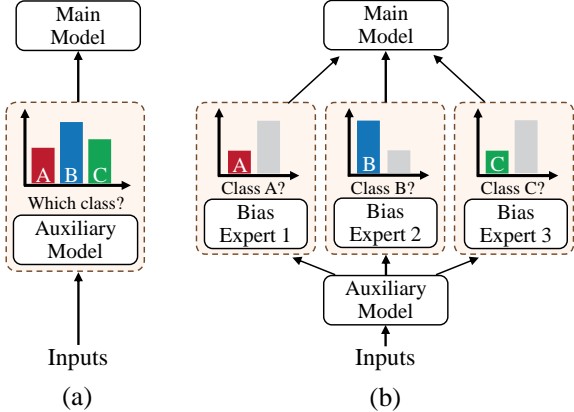

Figure 1: Comparison between (a) existing debiasing methods and (b) our proposed framework. (a): identify biased examples by using an auxiliary model trained with the multi-class learning objective, the same as the main model. (b): introduce intermediate binary classifiers, called bias experts, between the auxiliary model and the main model.

work well on a specific data distribution (Ribeiro et al., 2016; Zhu et al., 2017; Gururangan et al., 2018; McCoy et al., 2019). For instance, in natural language inference (NLI) tasks, neural networks are more likely to predict examples containing negation words as the contradiction class (Gururangan et al., 2018; Poliak et al., 2018). Ultimately, this unintended usage of bias results in poor model performance on out-of-distribution data or bias-conflicting[2] examples, where spurious correlations do not exist. Therefore, it is important to develop debiasing methods.

Previous works have addressed the issue by guiding the main model to down-weight biased examples during training, which are identified by auxiliary models. The auxiliary models are models that intentionally learn bias features by utilizing human supervision (Kim et al., 2019; Schuster et al., 2019; Clark et al., 2019; Mahabadi et al., 2020) or

---

[*]These authors contributed equally to this work.
[1]Our code is available at https://github.com/jej127/Bias-Experts.

[2]Bias-conflicting examples indicate examples that cannot be correctly predicted by solely relying on biases.

prior knowledge about dataset biases (Bahng et al., 2020). However, acquiring human supervision or prior knowledge about biases in myriad datasets is a dauntingly laborious and costly process. Recent studies have therefore shifted towards training the auxiliary models without human guidance (or bias labels). These studies attempt to make the auxiliary model biased by limiting the model capacity (Sanh et al., 2021), training for fewer epochs (Liu et al., 2021), restricting accessible data (Utama et al., 2020b; Kim et al., 2022), or using GCE loss (Nam et al., 2020).

Despite promising results, they still have room for further improvement. Specifically, the multi-class learning objective (e.g., softmax cross-entropy), which previous works have naively used to train the auxiliary model, may harm the bias identification ability of the model from a few aspects: a regularization effect and a competitive nature across classes. Feldman et al. (2019) have demonstrated that the multi-class learning objective has a regularization effect that prevents the model from overfitting on the training data. This is generally desirable for training the model, but not for training the auxiliary model that needs to be overfitted to bias features. Since the softmax normalization results in the sum of all the class logits equal to one, lowering the prediction confidence in one class necessarily increases the prediction confidence in the other classes (Wen et al., 2022). It can easily lead to overconfident predictions and may cause the auxiliary model to misidentify unbiased examples as biased examples.

To address the aforementioned problem, we propose a novel debiasing framework that introduces bias experts between the auxiliary model and the main model. As shown in Figure 1, the bias expert is a binary classifier that identifies biased examples within a specific class rather than multiple classes. In practice, we first cast the multi-class classification task into multiple individual binary classification tasks via the One-vs-Rest approach (Rifkin and Klautau, 2004), which encourages the model to individually learn the bias attributes of the target class without the influence of other classes. In addition, our framework motivated by Nam et al. (2020) highlights biased examples of target classes that the auxiliary model deems "easy", allowing bias experts to focus more on learning the bias attributes of the class in charge. Finally, we train the main model by reweighting the loss of the exam-

ples with bias experts.

We validate the effectiveness of the proposed framework on various challenging datasets. Experiment results show that our framework significantly improves performance on all datasets. The contributions of this paper are as follows:

- We show that the multi-class learning objective may harm the bias identification ability of the auxiliary model.

- We propose a novel debiasing approach based on bias experts to improve the out-of-distribution performance of existing debiasing methods.

- Through various empirical evaluations, we show that introducing bias experts improves debiasing performance.

## 2  Motivation

In this section, we describe our empirical observations on the existing debiasing approach. These observations serve as intuitions for designing and understanding our bias experts. We first provide a description of the experimental setup in Section 2.1. Then we provide our empirical observations in Section 2.2.

### 2.1  Setup

Consider a training dataset $\mathcal{D}$ of a task where each input $x \in \mathcal{X}$ is classified to a label $y \in \mathcal{Y}$. Then each input $x$ can be represented by a set of attributes $A_x = \{a_1, ..., a_k\}$. The set $A_x$ includes target attributes $a_t \in T$, which are expected to be learned to perform the target task. It also includes bias attributes $a_b \in B$, which have a high correlation with $y$ within a data distribution $\mathcal{D}$, but such correlation does not hold in other distributions.

In debiasing methods, the auxiliary model assigns different weights to biased examples and bias-conflicting examples. These works classify examples that can be correctly predicted based solely on $a_b$ as biased examples and regard examples that do not fall into this category as bias-conflicting examples. To achieve this, an intentionally biased classifier $f_b : \mathcal{X} \rightarrow \mathcal{Y}$ is employed as the auxiliary model. They identify biased examples by $f_b$, which primarily rely on bias attributes $a_b$, and down-weight them when training the main model $f_d : \mathcal{X} \rightarrow \mathcal{Y}$. Namely, they consider examples that are correctly predicted by $f_b$ as biased examples,

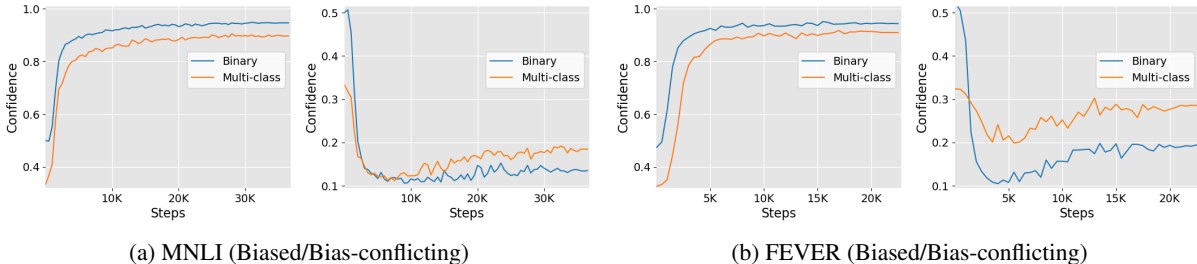

| (a) MNLI (Biased/Bias-conflicting) | (b) FEVER (Biased/Bias-conflicting) |

Figure 2: The process of a neural network learning on the biased datasets. For each dataset, the left and the right plots correspond to the result on biased and bias-conflicting example groups. Confidence refers to the average prediction probability of the correct class on the examples.

and down-weight their importance when training the main model.

## 2.2 Observations

In the aforementioned setup, we observe the models trained with two types of objective functions: multi-class learning objective and binary learning objective. The observations are made on the MNLI and FEVER datasets, which are widely studied in this area.

**Bias attributes are learned slower** Our first observation is that a model trained with the multi-class learning objective learns bias attributes more slowly. As shown in Figure 2, confidence on biased examples of the model trained with the multi-class learning objective increases more slowly than those trained with the binary learning objective. Moreover, even after convergence, it shows lower confidence on biased examples.

**Target attributes are learned faster** The second observation is that the model trained with the multi-class learning objective learns the target attributes faster. Figure 2 reports the average confidence of models on bias-conflicting examples. To predict bias-conflicting examples correctly, the models have to leverage $a_t$ rather than $a_b$. As seen in the figure, the confidence on bias-conflicting examples of the model trained with the multi-class learning objective starts to increase ahead of the model trained on the binary learning objective. In addition, the degree of confidence increment is also larger in the multi-class learning objective.

**Multi-class learning objective is sub-optimal** From these observations, we argue that the multi-class learning objective is sub-optimal to training the auxiliary model. Existing debiasing methods typically train the auxiliary model with the multi-class learning objective and consider examples that

the auxiliary model correctly predicts with high confidence as biased examples. Therefore, it should be able to have high confidence on biased examples and low confidence on bias-conflicting examples. But, as we observed before, the model trained with the multi-class learning objective shows the opposite tendency. These observations have interesting connections with those of Feldman et al. (2019). They analyze that multi-class classification reduces overfitting due to remaining uncertainties across multiple classes. Looking at this from the perspective of a biased model, it is similar to our results in that the multi-class learning objective can hinder the auxiliary model from learning bias attributes in biased examples.

## 3 Proposed method

Based on our observation in Section 2, we propose a novel debiasing framework, introducing bias experts between the auxiliary model and the main model to improve bias mitigation methods. It first splits the $k$-class classification dataset $\mathcal{D} = \bigcup_{i=1}^{k} D_i$ into multiple binary classification datasets $\mathcal{B}_1, \mathcal{B}_2, ..., \mathcal{B}_k$, where $\mathcal{D}_i = \{(x,y)|y=i\}$, $\mathcal{B}_i = \{(x, \mathbb{1}_{y=i})|(x,y) \in \mathcal{D})\}$, and $\mathbb{1}$ is an indicator function. Then for each $i \in \{1, ..., k\}$, we train the bias expert $f_b^i : \mathcal{X} \to \{0, 1\}$ on $\mathcal{B}_i$, with a weighted loss $\mathcal{L}_i$ to amplify the bias of $f_b^i$. While we need to train more bias experts as the number of classes (i.e., $k$) increases, most NLU tasks have a moderate number of classes[3]. Therefore, our method is applicable across various tasks, including NLI, paraphrase identification, sentiment analysis, and topic classification. Finally, we train the debiased model $f_d$ by reweighting the loss of each example with $\{f_b^1, ..., f_b^k\}$. The overall process of our

---

[3]For instance, there are typically 2-3 classes in NLI, 2 classes in paraphrase identification, 2-5 classes in sentiment analysis, and 4-20 classes in topic classification.

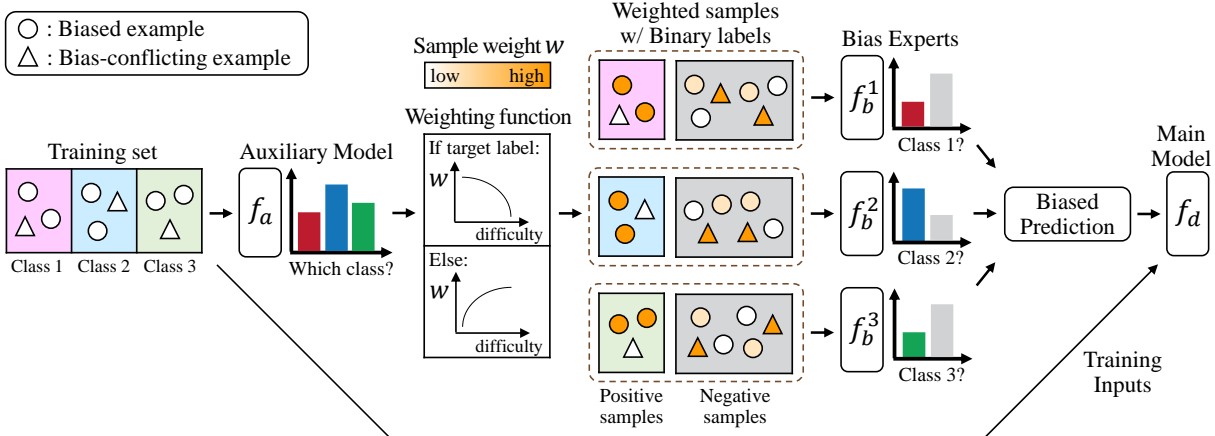

Figure 3: The overall pipeline of our method. Each bias expert is trained by assigning high weights to biased examples belonging to the target class and bias-conflicting examples not belonging to the target class. As a result, each bias expert focuses on biased examples of its target class.

framework is illustrated conceptually in Figure 3.

## 3.1 One-vs-Rest

In Section 2, we observe that the multi-class learning objective impedes the auxiliary model from being biased. Thus, we introduce a One-vs-Rest (OvR, also called One-vs-All) approach to train the auxiliary model on the binary learning objective.

The OvR is an approach in machine learning to making binary classification algorithms (e.g., logistic regression model) capable of working on multi-class classification dataset $\mathcal{D}$ by splitting $\mathcal{D}$ into multiple binary datasets $\mathcal{B}_1, \mathcal{B}_2, ..., \mathcal{B}_k$. For example, the MNLI dataset, which includes "contradiction", "entailment" and "neutral" classes, is divided into the following three binary classification datasets (target vs non-target) through the OvR approach:

- $\mathcal{B}_1$: Contradiction vs Non-Contradiction

- $\mathcal{B}_2$: Entailment vs Non-Entailment

- $\mathcal{B}_3$: Neutral vs Non-Neutral

This strategy enables the model to individually learn the bias attributes of the target class without the competition across different classes, which is caused by the softmax normalization.

## 3.2 Training bias experts

In this work, we define the bias expert $f_b$ to be a binary classifier that identifies biased examples within a specific class. We expect $f_b$ to learn the bias attributes of the target class and identify biased examples by relying on them. Specifically, the

bias expert is required to predict only the biased examples of the target class as the target class with high confidence, while, on bias-conflicting examples, the bias expert is required to predict with low confidence. However, by increasing the weight of all easy examples, the bias expert can learn the bias attributes of the non-target class and predict bias-conflicting examples as the target class with high confidence, which is not in line with our expectations. For example, in the case of the MNLI, the bias-conflicting example of a neutral class with neither lexical overlap nor negation words can be identified as biased with high confidence[4].

To discourage bias experts from learning bias attributes of non-target classes, we assign large weight to bias-conflicting examples of non-target classes. Therefore, inspired by Zhang and Sabuncu (2018) and Nam et al. (2020), we train bias experts by down-weighting the loss of biased examples in the non-target class based on the confidence of the auxiliary model as follows:

$$\mathcal{L}_{\text{non-target},i} = -\sum_{(x,y)\in\mathcal{D}_i^c} \frac{(1-q_{x,y})^\alpha \log(1-\sigma(f_b^i(x)))}{|\mathcal{D}_i^c|},$$
(1)

where $i$ is the target class, $q_{x,y}$ denotes the weight of the example $(x, y)$, and $\sigma(\cdot)$ is the sigmoid activation function. $\alpha$ is a hyperparameter that controls the degree of amplification. At the same time, we down-weight bias-conflicting exam-

---

[4]Lexical overlap and the presence of negation words are the bias attributes of entailment and contradiction classes, respectively.

ples of the target class:

$$\mathcal{L}_{\text{target},i} = - \sum_{(x,y)\in\mathcal{D}_i} \frac{q_{x,y}^{\alpha} \log \sigma(f_b^i(x))}{|\mathcal{D}_i|}. \quad (2)$$

We expect that these losses encourage bias experts to learn the bias attributes of the target class rather than those of the non-target class. Thus, we minimize them over $k$ bias experts:

$$\mathcal{L}(\theta_b) = \frac{1}{k} \sum_{i=1}^{k} (\lambda_1 \mathcal{L}_{\text{target},i} + \lambda_2 \mathcal{L}_{\text{non-target},i}), \quad (3)$$

where $\lambda_1$ and $\lambda_2$ are the hyperparameters that control the class imbalance aroused by OvR. At this time, backpropagation is not performed for the auxiliary model.

### 3.3   Training debiased model

We train a debiased model $f_d$ using product-of-experts (PoE) which is widely used in NLU debiasing research (Clark et al., 2019; Mahabadi et al., 2020; Sanh et al., 2021). In PoE, $f_d$ is trained in an ensemble by combining the softmax outputs of $f_d$ and $f_b$. The ensemble loss for each example is:

$$\mathcal{L}(\theta_d) = -y \cdot \log \text{softmax}(\log p_d + \log p_b). \quad (4)$$

With this loss function, examples that bias experts predict correctly are down-weighted, thereby discouraging $f_d$ from learning attributes exploited by bias experts. Since we use multiple bias experts, we apply softmax to the final outputs of bias experts and use this as $p_b$. During the training, we freeze the parameters of $f_b$.

## 4   Experiments

### 4.1   Evaluation Tasks

We evaluate our model and baselines on three natural language understanding tasks that have out-of-distribution evaluation sets: natural language inference, fact verification, and paraphrase identification. We use accuracy as the performance metric for each task. The detailed information for each task is as follows:

**Natural language inference**   Natural language inference (NLI) is the task of determining whether a premise entails, contradicts, or is neutral to a hypothesis given a sentence pair. We train our proposed model and baselines on the MNLI training set (Williams et al., 2018). For evaluation, we evaluate the in-distribution performance of the models on the MNLI validation set and the out-of-distribution performance on HANS (McCoy et al., 2019), an evaluation set designed to determine whether models have adopted three predefined syntactic heuristics such as lexical overlap.

**Fact verification**   Fact verification involves determining if the evidence supports or refutes a claim, or if it lacks sufficient information. We train our proposed model and baselines on the FEVER training set (Thorne et al., 2018). For evaluation, we evaluate the in-distribution performance of the models on the FEVER validation set and the out-of-distribution performance on FEVER Symmetric (Schuster et al., 2019), an evaluation set designed to test whether models rely on spurious cues in claims.

**Paraphrase identification**   Paraphrase identification is the task of detecting whether a given pair of questions is semantically equivalent. We use the QQP[5] dataset and divide this dataset into training and validation sets so that the validation set contains 5k examples, following Udomcharoenchaikit et al. (2022). We train our proposed model and baselines on the resulting QQP training set. For evaluation, we evaluate the in-distribution performance of the models on the QQP validation set and the out-of-distribution performance on PAWS (Zhang et al., 2019) to test whether models learn to exploit lexical overlap bias.

### 4.2   Baselines

We compare multiple baselines, which are generally classified into two categories:

**Methods using prior knowledge of bias:**   In such methods, auxiliary models are trained by using the hand-crafted features which indicate how words in one sentence are shared with another sentence (Clark et al., 2019; Mahabadi et al., 2020; Utama et al., 2020a), or whether a certain n-gram occurs in a sentence (Schuster et al., 2019).

**Methods without targeting a specific bias:**   This group of methods forces auxiliary models to learn bias attributes either by reducing the training dataset size (Utama et al., 2020b), or restricting the model capacity (Sanh et al., 2021; Ghaddar et al., 2021), without targeting a specific bias type.

---

[5]https://quoradata.quora.com/First-Quora-Dataset-Release-Question-Pairs

| Method | Known biases | MNLI | | | FEVER | | | QQP | | |
|---|---|---|---|---|---|---|---|---|---|---|
| | | dev | HANS | Gap | dev | symm. | Gap | dev | PAWS | Gap |
| BERT-base (Devlin et al., 2019) | ✗ | 84.5 | 62.4 | 22.1 | 85.6 | 63.1 | 22.5 | 91.0 | 33.5 | 57.5 |
| Reweighting (Clark et al., 2019) | ✓ | 83.5 | 69.2 | 14.3 | - | - | - | - | - | - |
| Reweighting (Schuster et al., 2019) | ✓ | - | - | - | 84.6 | 66.5 | 18.1 | - | - | - |
| PoE (Clark et al., 2019) | ✓ | 82.9 | 67.9 | 15.0 | - | - | - | - | - | - |
| Conf-reg (Utama et al., 2020a) | ✓ | 84.3 | 69.1 | 15.7 | 86.4 | 66.2 | 20.2 | - | - | - |
| Reweighting (Utama et al., 2020b) | ✗ | 82.3 | 69.7 | 12.6 | 87.1 | 65.5 | 21.6 | 85.2 | 57.4 | **27.8** |
| PoE (Utama et al., 2020b) | ✗ | 81.9 | 66.8 | 15.1 | 85.9 | 65.8 | 20.1 | 86.1 | 56.3 | 29.8 |
| PoE (Sanh et al., 2021) | ✗ | 83.3 | 67.9 | 15.4 | 84.8 | 65.7 | 19.1 | 88.0 | 46.4 | 41.6 |
| Conf-reg (Utama et al., 2020b) | ✗ | **84.3** | 67.1 | 17.2 | **87.6** | 66.0 | 21.6 | 89.0 | 43.0 | 46.0 |
| Self-Debiasing (Ghaddar et al., 2021) | ✗ | 83.2 | 71.2 | 12.0 | - | - | - | **90.2** | 46.5 | 43.7 |
| Bias Experts (ours) | ✗ | 82.7 | **72.6** | **10.1** | 85.6 | **68.1** | **17.5** | 86.8 | **58.1** | 28.7 |

Table 1: Performances of models evaluated on MNLI, FEVER, QQP, and their corresponding challenge test sets. We also report the difference between in- and out-of-distribution performances as Gap. The results of baselines using prior knowledge of biases are from the original paper. We mark the best and the second-best performance in **bold** and underline, respectively.

## 4.3 Implementation details

In all tasks, we employ the BERT-base model with 110M parameters for the main model and use the BERT-tiny (Turc et al., 2019) model with 4M parameters for the auxiliary model and the bias experts[6]. We use AdamW as the optimizer for both the main model and the biased model, with a batch size of 32 and 3 epochs of training. In order to train the main model, we use a coefficient of 0.3 for the cross-entropy loss and a coefficient of 1.0 for the PoE loss, following Sanh et al. (2021). For the NLI task, we set the learning rate to 3e-5 and $\alpha$ to 0.2. For the fact verification task, we set the learning rate to 2e-5 and the $\alpha$ to 0.01. For the paraphrase identification task, we set the learning rate to 2e-5 and the $\alpha$ to 0.3.

## 4.4 Main results

We report experimental results on three natural language understanding tasks in Table 1. Each result is the average of the scores across 5 different runs. We observe that our model consistently outperforms the state-of-the-art on three out-of-distribution evaluation sets. Specifically, our model shows a performance improvement of 1.4%p, 2.1%p, and 0.7%p in HANS, FEVER Symmetric, and PAWS over the best-performing model, respectively. These results indicate that our framework is effective in identifying biased examples regardless of the type of the dataset. In addition, the gap between in- and out-of-distribution performances of our model is much

[6] https://github.com/huggingface/transformers

| Ablative Setting | MNLI | HANS | | |
|---|---|---|---|---|
| | | Ent | NEnt | Avg |
| Bias Experts (ours) | 82.7 | 93.5 | **51.8** | **72.6** |
| (1) w/o Amp. | 82.8 | **94.9** | 45.0 | 70.0 |
| (2) w/o OvR | 83.0 | 94.4 | 42.7 | 68.5 |
| (3) w/o Amp. and OvR | **83.1** | 94.5 | 41.4 | 67.9 |

Table 2: Ablation studies using accuracy on MNLI and HANS. We mark the best performance in **bold**.

smaller than the other methods, suggesting that our framework achieves more out-of-distribution performance gains with less in-distribution performance losses. On the other hand, we can observe that the proposed framework shows the lowest performance improvement in PAWS, the challenge test set of QQP. This is because only bias amplification affects the performance improvement since QQP is originally a binary problem.

## 4.5 Ablation study

**Effect of each module in our method** We compare different ablative settings of our method to demonstrate the effect of each part in our method. (1) w/o Amp. denotes omitting bias amplification of biased models by setting $\alpha = 0$ in Eq. 1 and Eq. 2, (2) w/o OvR denotes that we train the main model with an ensemble prediction of three auxiliary models trained with the multi-class learning objective, (3) w/o Amp. and OvR denotes that we remove both bias amplification and OvR approach. Table 2 indicates that all the components are

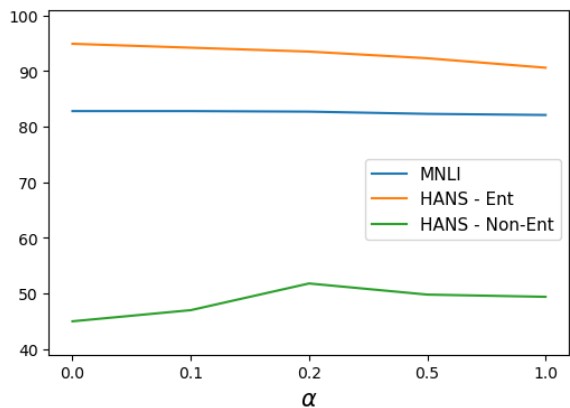

Figure 4: Performances of main models evaluated on MNLI and HANS with different $\alpha$.

| Method | MNLI | HANS |
|---|---|---|
| BERT-base | 84.5 | 62.4 |
| PoE (Utama et al., 2020b) | 81.9 | 66.8 |
| PoE (Sanh et al., 2021) | 83.3 | 67.9 |
| PoE + Ours ($T = 2k$) | **83.8** | 68.0 |
| PoE + Ours ($T = 10k$) | 83.3 | 71.0 |
| Bias Experts (ours) | 82.7 | **72.6** |

Table 3: Trade-off results between in- and out-of-distribution performances evaluated on MNLI and HANS. $T$ denotes the number of training steps for bias experts. We mark the best and the second-best performance in **bold** and underline, respectively.

important in the improvement of debiasing performance. Amplifying the target class bias of bias experts (w/o Amp.) is crucial to boost the out-of-distribution performance, indicating that focusing on learning biased examples of the target class improves the bias experts' ability to identify biased examples. In addition, omitting the OvR approach (w/o OvR) further decreases the performance, which suggests that training bias experts with the binary learning objective is key to improving the performance of bias experts. Furthermore, using both OvR and Amp. causes larger improvement in out-of-distribution performance compared to using one of them. These results show that the two components work complementary to each other, improving different aspects of bias experts. Additional results on the fact verification task are provided in Appendix A.6.

**Degree of bias amplification**    We vary the degree of bias amplification for the target class (i.e., $\alpha$) and compare the performances of the corresponding main models. Figure 4 shows the results. We first observe that debiasing with $\alpha = 0$ shows the lowest accuracy. This indicates that bias amplification helps bias experts focus on the target class bias, leading to better out-of-distribution performances of the main model. On the other hand, as $\alpha$ exceeds 0.2, both in-distribution and out-of-distribution performances start to decrease. We speculate that this is because the main model receives a smaller gradient update as $\alpha$ increases. This indicates that we should train the bias experts either by using a moderate level of $\alpha$ or by increasing the learning rate.

### 4.6   In- and Out-of-distribution Performances Trade-off

To verify that our method achieves a better trade-off between in- and out-of-distribution performances than baselines, we conduct an analysis by tuning the number of training steps for bias experts. The intuition behind tuning the number of training steps is that the biased models with smaller training steps are likely to less focus on bias attributes, resulting in a main model with higher in-distribution performance, compared to the setting where biased models are fully fine-tuned (e.g., for 3 epochs as in Section 4.3). The results are presented in Table 3. By comparing the results of PoE + Ours ($T = 2k$) in row 4 with those of Sanh et al. (2021) in row 3, where $T$ denotes the number of training steps for bias experts, we can see that our model can achieve the similar out-of-distribution performance gain in HANS with the smaller in-distribution performance degradation in MNLI. In addition, the comparison results between PoE + Ours ($T = 10k$) in row 5 and the method in Sanh et al. (2021) demonstrate that our model improves the out-of-distribution performance while preserving the in-distribution performance. These results indicate that our method facilitates a better balance between in- and out-of-distribution performances.

### 4.7   Analysis on Model Confidence

To further investigate the effectiveness of our bias experts, we use the kernel density estimation to draw the distributions of model confidence on two groups of examples: biased and bias-conflicting examples. Specifically, we experiment on the MNLI training set and compare our bias experts with the weak learner used in Sanh et al. (2021). The result is illustrated in Figure 5. It is observed that our bias

| Method | Ent | | | NEnt | | |
|---|---|---|---|---|---|---|
| | Lexical | Subseq | Const | Lexical | Subseq | Const |
| BERT-base | 99.5 | 99.2 | 99.4 | 52.4 | 10.0 | 17.8 |
| PoE (Sanh et al., 2021) | **90.2** | **97.0** | **96.3** | 66.0 | 18.0 | 40.1 |
| Bias Experts (ours) | 88.4 | 96.6 | 95.5 | **83.7** | **26.4** | **45.2** |

Table 4: Performances of main models evaluated on HANS for each heuristic. The columns Lexical, Subseq, and Const mean lexical overlap, subsequence, and constituency, respectively. We mark the best performance in **bold**.

| Method | BAR | NICO |
|---|---|---|
| ERM | 35.3 | 42.6 |
| ReBias (Bahng et al., 2020) | 37.0 | 45.2 |
| LfF (Nam et al., 2020) | 48.2 | 40.2 |
| LWBC (Kim et al., 2022) | 62.0 | 52.8 |
| Bias Experts (ours) | **64.2** | **54.1** |

Table 5: Performances of models evaluated on BAR and NICO. We compare our method with baselines that do not use bias labels. We mark the best and the second-best performance in **bold** and underline, respectively.

experts show higher confidence on biased examples and lower confidence on bias-conflicting examples on average, compared to the weak learner in Sanh et al. (2021). This indicates that the bias experts identify biased and bias-conflicting examples more precisely than the biased model applied in the previous work. As a result, we can observe in Table 4 that the main model trained in an ensemble with the bias experts achieves significant improvement over the baselines on the non-entailment subset of HANS, for all three heuristics, with only a small degradation on the entailment subset of HANS.

## 4.8 Experiment on Image Classification

We conduct experiments on computer vision tasks to verify that our framework is model-agnostic, and can be applied to other debiasing algorithms. For computer vision tasks, we apply bias experts to the bias committee algorithm (Kim et al., 2022). We experiment on two image classification datasets: 1) Biased Action Recognition (BAR): a real-world image dataset crafted to contain examples with spurious correlations between human action and place, and 2) NICO: a real-world dataset for simulating out-of-distribution image classification scenarios. We compare our model with baselines without explicit supervision on bias: ReBias (Bahng et al., 2020), LfF (Nam et al., 2020), and LWBC (Kim et al., 2022). Table 5 shows the results on both datasets. Our model achieves performance improve-

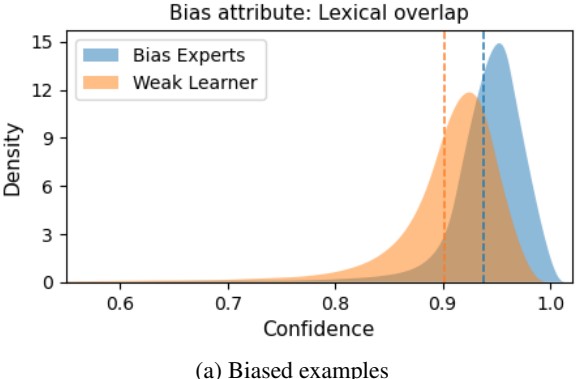

(a) Biased examples

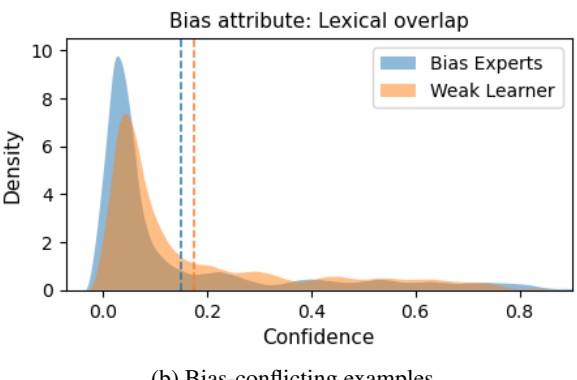

(b) Bias-conflicting examples

Figure 5: The kernel density estimation of models' confidence on (a) biased examples and (b) bias-conflicting examples in MNLI. The weak learner is the auxiliary model used in Sanh et al. (2021). Each plot includes dashed lines representing the average confidence of each model.

ments of 2.2%p and 1.3%p in BAR and NICO over LWBC, respectively. The results show that our framework can be used complementary with other debiasing algorithms in computer vision tasks.

## 5 Related Work

### 5.1 Debiasing with prior knowledge of bias

Earlier bias mitigation methods typically combine human prior knowledge with data-driven training. They define a specific bias type in advance and train the auxiliary model to learn the predefined bias

type. For example, the auxiliary model is trained to learn a word overlap bias or a hypothesis-only bias (He et al., 2019; Mahabadi et al., 2020) in NLI, a question-only bias (Cadène et al., 2019) in VQA, and a texture bias (Bahng et al., 2020) in image classification. Then, the auxiliary model reduces the importance of biased examples when training the main model by reweighting examples (Schuster et al., 2019), regularizing the main model confidence (Utama et al., 2020a; Mahabadi et al., 2020), or combining with the main model in the PoE manner (Clark et al., 2019). As a result, the main model focuses more on bias-conflicting examples in which exploiting solely bias attributes is not sufficient to make correct predictions. While these methods provide a remarkable improvement in the out-of-distribution performance, manually identifying all bias types in tremendous datasets is a time-consuming and costly process.

## 5.2 Debiasing without targeting a specific bias

To address these limitations, several attempts were conducted to train the auxiliary model without human intervention. These works force the auxiliary model to learn the bias attributes in datasets by restricting the size of training data (Utama et al., 2020b), limiting the capacity of the model (Bras et al., 2020; Clark et al., 2020; Sanh et al., 2021; Yaghoobzadeh et al., 2021; Ghaddar et al., 2021), training the model with GCE loss (Nam et al., 2020), or reducing the training epochs (Liu et al., 2021). Then, they consider the examples for which the auxiliary model struggles to predict the correct class as bias-conflicting ones and assign large weights to them when training the main model. Recently, to further improve debiasing methods, Kim et al. (2022) use the consensus of multiple auxiliary models to identify biased examples and determine their weights, Ahn et al. (2023) use the norm of the sample gradient to determine the importance of each sample, Lee et al. (2021) augment bias-conflicting features with feature-mixing techniques, and Lee et al. (2023) propose a method which removes the bias-conflicting examples from a training set to amplify bias for a biased model. Lyu et al. (2023) take a contrastive learning approach to mitigate bias features and incorporate the dynamic effects of biases. Although these prior efforts achieve promising results, to our knowledge, none of them attempted to improve the effectiveness of debiasing methods by tackling a detrimental

effect of the multi-class learning objective on the bias identification ability of an auxiliary model.

## 6 Conclusion

We have proposed a novel framework that introduces a binary classifier between the main model and the auxiliary model, called bias experts, for improving the bias mitigation method. The main intuition of this work is to mitigate the incorrect bias identification caused by the naive application of the multi-class learning objective to train the auxiliary model. We train our bias experts via the One-vs-Rest approach, pushing each bias expert to focus more on the biased examples in its target class. Through various experiments and ablation studies, we demonstrate our framework effectively alleviates the aforementioned problem and improves existing bias mitigation methods.

## Limitations

Although we have demonstrated the efficacy of our method in improving bias mitigation, there are two limitations that should be addressed in the future:

(1) Since we have to train individual binary classifiers for each class, a limitation of our work is that it may lead to large memory usage as the number of classes increases. However, to the best of our knowledge, most NLU tasks consist of only a moderate number of classes. For example, we found that there are typically 2-3 classes in NLI, 2 classes in paraphrase identification, 2-5 classes in sentiment analysis, and 4-20 classes in topic classification. In addition, our work has validated wide applicability in datasets with a moderate number of classes across various NLU and image classification tasks. In the future, we plan to alleviate this limitation by exploring the way to adopt the parameter-efficient fine-tuning method for training bias experts.

(2) We introduce hyperparameters in our work, which could be problematic in debiasing works since most out-of-distribution datasets do not provide a validation set for tuning hyperparameters, as noted by Utama et al. (2020a). With this in mind, we have sought to minimize the number of hyperparameters additionally introduced in our work. Specifically, there are three additionally introduced hyperparameters, $\lambda_1$, $\lambda_2$, and $\alpha$. Following Wen et al. (2022), we set $\lambda_1 = (k-1)/k$ and $\lambda_2 = 1/k$, where $k$ denotes the number of classes. Thus, $\lambda_1$ and $\lambda_2$ are not involved in the tuning process. In

addition, compared with previous works, the performance of the proposed method is less sensitive to the hyperparameter $\alpha$. Specifically, in HANS, the performance of our method varies with a variance of 0.99, depending on the value of $\alpha$. This variance is lower than 1.10 - 8.80, the variances of performance observed in prior works (Utama et al., 2020b; Sanh et al., 2021; Kim et al., 2022; Lyu et al., 2023). In the future, we plan to investigate how to tune hyperparameters without leaking information about the bias.

## Ethics Statement

We acknowledge that the high out-of-distribution performance of our method is achieved at the expense of the in-distribution performance (i.e., at the expense of the performance in the majority group). One of the risks that can stem from this trade-off is that errors in the majority group could increase in applications such as toxicity classification, facial recognition, and medical imaging, potentially leading to reverse discrimination against samples in such a majority group.

## Acknowledgements

This work was supported by the Basic Research Program through the National Research Foundation of Korea (NRF) grant funded by the Korea government (MSIT) (2021R1A2C3010430) and Institute of Information & Communications Technology Planning & Evaluation (IITP) grant funded by the Korea government (MSIT) (No.2019-0-00079, Artificial Intelligence Graduate School Program (Korea University)).

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

# A   Appendix

## A.1   Dataset Statistics

Table 6 shows data statistics for the datasets used in our experiments.

|    | Dataset | #samples | #classes |
|----|---------|----------|----------|
| ID | MNLI    | 401k     | 3        |
|    | FEVER   | 258k     | 3        |
|    | QQP     | 399k     | 2        |
| OOD | HANS   | 30k      | 2        |
|    | FEVER Symm. | 712  | 3        |
|    | PAWS    | 677      | 2        |

Table 6: Detailed dataset statistics. ID and OOD denote in-distribution and out-of-distribution data, respectively.

## A.2   Details of Preliminary Experiments

In this section, we present the details of the preliminary experiments discussed in Section 2. We collect biased/bias-conflicting examples from the training set of MNLI and FEVER. For MNLI, we select examples where all the hypothesis words occur in the premise in terms of exact match. Among those selected examples, we classify examples with the entailment class as biased and others as bias-conflicting. For FEVER, we select examples where any of the Top-10 LMI-ranked bigrams, listed in Schuster et al. (2019), appear in the claim also in terms of exact match. We then classify the examples with the refute class as biased and others as bias-conflicting.

## A.3   Effect of merging method

We use as many bias experts as the number of classes in multi-class classification, thus we need to merge their predictions into one probability distribution to train the main model in a PoE manner. We consider two different strategies for merging predictions from different bias experts: (1) softmax and (2) softplus. From Table 7, we observe that both strategies show higher out-of-distribution performances compared to the vanilla PoE baseline. In particular, merging with softmax outperforms merging with softplus: 72.6% vs 69.6% on HANS. This suggests that applying softmax is best suited for our debiasing method.

## A.4   Handling class imbalance

The OvR approach we introduced causes an imbalance between classes because it classifies all

| Merging Method | MNLI | HANS | | |
|----------------|------|------|------|------|
|                |      | Ent  | NEnt | Avg  |
| PoE (Sanh et al., 2021) | **83.1** | **94.5** | 41.4 | 67.9 |
| PoE (Utama et al., 2020b) | 81.9 | 90.9 | 42.8 | 66.8 |
| (1) Softmax | 82.7 | 93.5 | **51.8** | **72.6** |
| (2) Softplus | 82.7 | 94.0 | 45.2 | 69.6 |

Table 7: Performances of main models evaluated on MNLI and HANS with different strategies for merging expert predictions. We mark the best performance in **bold**.

| Balancing Method | MNLI | HANS | | |
|------------------|------|------|------|------|
|                  |      | Ent  | NEnt | Avg  |
| (1) Reweighting | **82.7** | 93.5 | **51.8** | **72.6** |
| (2) Over-sampling | 82.7 | **95.3** | 46.6 | 71.0 |
| (3) Under-sampling | 81.7 | 94.5 | 48.0 | 71.2 |
| (4) w/o Balancing | 82.7 | 94.6 | 44.2 | 69.4 |

Table 8: Performances of main models evaluated on MNLI and HANS with different strategies for balancing positive/negative samples. We mark the best performance in **bold**.

classes except the target class as a non-target class. In this analysis, to alleviate this problem, we compare three widely used strategies for mitigating the imbalance class problem: (1) reweighting, (2) over-sampling, and (3) under-sampling. Table 8 shows the accuracy of the main models on MNLI and HANS. All three methods, the reweighting, over-sampling, and under-sampling methods show better performance than the method without applying the imbalance mitigation method (i.e., w/o Balancing). Also, when using reweighting, the performance of the main model is the highest.

## A.5   Qualitative Results

Table 9 illustrates examples to show how our proposed framework improves the identification performance of biased examples. The examples show that bias experts trained with the binary learning objective make predictions with higher confidence, compared to the weak learner in Sanh et al. (2021), the biased model trained with the multi-class learning objective. This indicates that each bias expert individually learns the bias attributes (e.g., high lexical overlap, or the presence of negation words such as "no" and "never") of the target class without being affected by the confidence of other classes.

| Bias Attribute | Premise | Hypothesis | Class | Weak Learner | Bias Experts |
|---|---|---|---|---|---|
| Lexical overlap | We have Kroger but not a Skaggs. | We have a Kroger but no Skaggs. | Contradiction | 18.9% | 7.2% |
| | | | **Entailment** | 75.6% | **91.4%** |
| | | | Neutral | 5.5% | 1.4% |
| | And no surprise, the bugger was already inside. | The bugger was inside. | Contradiction | 28.9% | 17.2% |
| | | | **Entailment** | 66.2% | **81.5%** |
| | | | Neutral | 4.9% | 1.3% |
| Presence of Negation | I didn't hear what Mr. Inglethorp replied. | Mr. Inglethorp never said anything back. | **Contradiction** | 64.2% | **83.6%** |
| | | | Entailment | 5.3% | 1.6% |
| | | | Neutral | 30.5% | 14.8% |
| | That's the only way to keep you from being punished. | There's no way to keep you from being punished. | **Contradiction** | 65.4% | **86.2%** |
| | | | Entailment | 29.7% | 12.7% |
| | | | Neutral | 4.9% | 1.1% |

Table 9: Qualitative comparison of the models' confidence. We compare our bias experts with the weak learner used in Sanh et al. (2021). The bold class denotes the true label.

| Ablative Setting | FEVER | |
|---|---|---|
| | dev | symm. |
| Bias Experts (ours) | **85.6** | **68.1** |
| (1) w/o Amp. | 85.1 | 66.9 |
| (2) w/o OvR | 84.9 | 66.0 |
| (3) w/o Amp. and OvR | 84.8 | 65.7 |

Table 10: Ablation studies using accuracy on FEVER and FEVER Symmetric. We mark the best performance in **bold**.

## A.6 Additional Ablation on Fact Verification

To further study the efficacy of each component in our method, we conduct additional ablation studies on FEVER and FEVER Symmetric datasets. The results are shown in Table 10, and the ablative settings are the same as those in Section 4.5. As shown in the table, using both OvR and Amp. obtains the best out-of-distribution performance, demonstrating the importance of both components. In addition, OvR contributes more to improving the out-of-distribution performance, compared to bias amplification. These results are consistent with those reported in Table 2.