# OpenReview forum: "Improving Bias Mitigation through Bias Experts in Natural Language Understanding"
_EMNLP/2023/Conference — EMNLP 2023 Main_

### Official Review · Reviewer_R64f · 2023-08-03

**Soundness:** 3

**Excitement:**

3: Ambivalent: It has merits (e.g., it reports state-of-the-art results, the idea is nice), but there are key weaknesses (e.g., it describes incremental work), and it can significantly benefit from another round of revision. However, I won't object to accepting it if my co-reviewers champion it.

**Missing References:**

"Increasing Robustness to Spurious Correlations using Forgettable Examples" Yaghoobzadeh et al.

"Learning to Model and Ignore Dataset Bias with Mixed Capacity Ensembles" Clark et al.

There is a lot of additional work on de-biasing that has focused on image classification or VQA that is not cited, I don't think they need to be comprehensively covered but I think the covering them in a bit more detail would be good. Some examples:

"Revisiting the Importance of Amplifying Bias for Debiasing" Lee at al.

"EnD: Entangling and Disentangling deep representations for bias correction" Tartaglione et al.


**Paper Topic And Main Contributions:**

This paper consider the problem of de-biasing a model without prior knowledge of the bias. Similar to other works they train a bias-only model the reflects the bias and us it to de-biasing another model. They propose using a one-vs-all classifier instead of a multi-class classifier, as well as a bias amplification re-weighting scheme, will result in a bias-only model that better captures the bias.

**Questions For The Authors:**

Any reason to not have the auxiliary model used to get the weights for 3.2 not be also be one-vs-all?

What model is being used for the image classification experiments? The appendix at least should include some additional details.

The explanation of why one vs. all bias-only models are better in 080-090 was not very clear to me. Presumably it is due to some kind of regularizing effect in multi-class models that reduces bias but I don't quite have an intuition on what that would be. Is there any other intuition that would be make things more clear?

**Reasons To Accept:**

- Evaluate on many well studied de-biasing tasks with positive results
- Section 2 helps motivate the method by providing empirical evidence for the hypothesis that the multi-class bias-only model are worse at capturing bias then one-vs-rest models.
- Ablation study helps verify the value of the individual components.

**Reasons To Reject:**

- Introduces several hyper-parameters, which is always problematic in de-biasing works since one shouldn't assume there is a validation set of non-biased to hyper-parameters since using such a set would "leak" information about the bias. The work also does not discuss how the parameters are tuned for their experiments.
- I am left a little uncertain how much the bias amplification vs. the OvR is contributing, I see the authors even report SoTA on QQP despite it being a binary problem. The is particularly important since I understand the bias amplification part mostly follows prior work. The ablation helps but it is only on HANS which can be a pretty unreliable eval set (e.g. "BERTs of a feather do not generalize together: Large variability in
generalization across models with similar test set performance"  McCoy, et al.), ablation results on the other other tasks would help.
- Overall seems like a more incremental change relative prior work
- The evaluation approach using BERT-Base is what was used in prior work, but is starting to feel like a very out-of-date setup.

**Reproducibility:**

3: Could reproduce the results with some difficulty. The settings of parameters are underspecified or subjectively determined; the training/evaluation data are not widely available.

**Reviewer Confidence:**

4: Quite sure. I tried to check the important points carefully. It's unlikely, though conceivable, that I missed something that should affect my ratings.

**Typos Grammar Style And Presentation Improvements:**

I know from the cited works that the idea in 270-300 is to amplify the bias, but the explanation in 3.2 appears to be completely circular since it reads like one should use knowledge of the bias to discover the bias.

---

> ### Author Rebuttal · Authors · 2023-08-29
>
> **Reason to reject 1.** Introduces several hyper-parameters, which is always problematic in de-biasing works since one shouldn't assume there is a validation set of non-biased to hyper-parameters since using such a set would "leak" information about the bias. The work also does not discuss how the parameters are tuned for their experiments.
>
> **Answer.** Yes, we agree that introducing several hyper-parameters is problematic in debiasing works. With this in mind, we have sought to minimize the number of hyper-parameters in our work. Specifically, there are three additionally introduced hyper-parameters, $\lambda_{1}$,  $\lambda_{2}$, and $\alpha$. We have introduced $\lambda_{1}$ and $\lambda_{2}$ to mitigate the class imbalance problem aroused by OvR. Following [1], we set $\lambda_{1}=(k-1)/k$ and $\lambda_{2}=1/k$, where $k$ denotes the number of classes. Thus, $\lambda_{1}$ and $\lambda_{2}$ are not involved in the tuning process.
>
> In addition, compared with previous works, the performance of the proposed method is less sensitive to the hyperparameter $\alpha$ introduced to control the degree of bias amplification. Specifically, in HANS, the performance of our method varies from 70.0 to 72.7 (±0.99) depending on the hyperparameter value. On the other hand, the performance of [2] ranges from 66.5 to 72.0 (±1.70, Figure 5), the performance of [3] ranges from 64.9 to 68.3 (±1.10, Table 6) depending on the hyperparameter value.  In addition, [4] shows that the performance of the model varies from 34.0 to 57.5 in PAWS (±8.80, Figure 4), and the performance of [5] varies from 77.5 to 85.5 in CelebA (±2.60, Figure 3) depending on the hyperparameters.
>
> However, we still agree that introducing hyper-parameters in debiasing works, where most out-of-distribution datasets do not provide a validation set, is problematic. Therefore, in the future, we plan to investigate how to tune hyper-parameters without leaking information about the bias. We will also add a discussion regarding this issue in the final version.
>
> [1] Wen et al., SphereFace2: Binary Classification is All You Need for Deep Face Recognition, ICLR 2022. \
> [2] Sanh et al., Learning from others' mistakes: Avoiding dataset biases without modeling them, ICLR 2021. \
> [3] Lyu, et al., Feature-level debiased natural language understanding, AAAI 2023. \
> [4] Utama et al., Towards Debiasing NLU Models from Unknown Biases, EMNLP 2020. \
> [5] Kim et al., Learning Debiased Classifier with Biased Committee, NeurIPS 2022.
>
> &nbsp;
>
> **Reason to reject 2.** I am left a little uncertain how much the bias amplification vs. the OvR is contributing. The ablation helps but it is only on HANS which can be a pretty unreliable evaluation set. Ablation results on the other other tasks would help.
>
> **Answer.** Thanks for the constructive comment. Therefore, based on your comment, we conducted an additional ablation experiment on FEVER and attached the results as below for presenting more reliable results. As shown in the table, OvR consistently contributes more significantly to the OOD performance on both commonly used evaluation dataset in the debiasing domain. We will include these results in the final version.
>
> |   Ablative Setting  | FEVER | FEVER-Symmetric | MNLI | HANS  |
> |-------------------|:-----:|:-----:|:------:|:------:|
> | Bias Experts (ours) |  85.6 |  **68.1** | 82.7 | **72.6** |
> | w/o Amp.            |  85.1 |  66.9 | 82.8 | 70.0 |
> | w/o OvR             |  84.9 |  66.0 | 83.0 | 68.5 |
> | w/o Amp. and OvR    |  84.8 |  65.7 | 83.1 | 67.9 |
>
> &nbsp;
>
> **Reason to reject 3.** Overall seems like a more incremental change relative prior work.
>
> **Answer.** We believe that our work has brought significant contribution to the debiasing domain. To the best of our knowledge, while most of the previous works have been studied to improve the way in utilizing the trained biased model, the way on how to effectively train a biased model remains under-explored. That being said, they generally follow the training strategies proposed in [6,7]. Meanwhile, our work presents a novel and effective way to train the biased model and has achieved significant performance improvement on various challenging datasets.
>
> [6] Sanh et al., Learning from others' mistakes: Avoiding dataset biases without modeling them, ICLR 2021.\
> [7] Utama et al., Towards Debiasing NLU Models from Unknown Biases, EMNLP 2020.
>
> &nbsp;
>
> **Reason to reject 4.** The evaluation approach using BERT-base is what was used in prior work, but is starting to feel like a very out-of-date setup.
>
> **Answer.** We have performed experiments on the BERT-base for comparison with previous works.
>
> &nbsp;
>
> **Question 1.** Any reason to not have the auxiliary model used to get the weights for 3.2 not also be one-vs-all?
>
> **Answer.** Our design choice of the auxiliary model is selected by the empirical observation that the change of performances of the main model is negligible when the one-vs-all approach is applied to train the auxiliary model. Specifically, when one-vs-all is applied to the auxiliary model, the MNLI performance has increased from 82.7 to 82.9, while the HANS performances have remained the same, 72.6. Thus, we adopt a multi-class classifier as an auxiliary model, which requires relatively less memory.
>
> Besides, in the case of using the probability of each bias expert itself for amplifying biased examples, the variance of the performance has been largely increased (1.47 compared to 0.77), while only showing slight improvement of the main model performance in the OOD data (73.0 compared to 72.6). Therefore, our methodology design uses the output (logits) of the auxiliary model, instead of the output from bias experts when calculating the weights.
>
> &nbsp;
>
> **Question 2.** What model is being used for the image classification experiments? The appendix at least should include some additional details.
>
> **Answer.** Thanks for your thoughtful feedback. We use ResNet-18 as a backbone network for the image classification experiments for a fair comparison with [8] and we replace the multi-class biased classifiers with biased experts. Specifically, in the BAR task, which has six classes, we assign five bias experts to each class, instead of using 30 multi-class classifiers. We will attach these details in the appendix.
>
> [8] Kim et al., Learning Debiased Classifier with Biased Committee, NeurIPS 2022.
>
> &nbsp;
>
> **Question 3.** The explanation of why one vs. all bias-only models are better in 080-090 was not very clear. Is there any other intuition that would make things more clear?
>
> **Answer.** To elaborate further, the multi-class learning objective (e.g., softmax cross-entropy) has several drawbacks in training biased classifiers: First of all, it has a competitive nature across classes [9]. In the multi-class learning objective, increasing the confidence of one class will necessarily decrease the confidence of some other classes, thus the model may predict bias-conflicting examples as biased when logits of other classes are relatively lower than the correct one. Secondly, as you mentioned above, it has a regularization effect. The multi-class learning objective could reduce the overfitting of the auxiliary model on the biased features due to the regularization effect [10]. Therefore, considering the aforementioned characteristics of the multi-class learning objectives, the one-vs-all approach is more suitable to train the biased model.
>
> [9] Wen et al., SphereFace2: Binary Classification is All You Need for Deep Face Recognition, ICLR 2022.\
> [10] Feldman et al., The advantages of multiple classes for reducing overfitting from test set reuse, ICML 2019.
>
> &nbsp;
>
> **Missing Reference.** We will update the related work section with the missing references you pointed out.

---

### Official Review · Reviewer_Vam3 · 2023-08-05

**Soundness:** 4

**Excitement:**

3: Ambivalent: It has merits (e.g., it reports state-of-the-art results, the idea is nice), but there are key weaknesses (e.g., it describes incremental work), and it can significantly benefit from another round of revision. However, I won't object to accepting it if my co-reviewers champion it.

**Missing References:**

[1] Ghaddar, A., Langlais, P., Rezagholizadeh, M., & Rashid, A. (2021). End-to-end self-debiasing framework for robust NLU training. ACL/IJCNLP (Findings) 2021

[2] Lyu, Yougang, et al. "Feature-level debiased natural language understanding." AAAI 2023

**Paper Topic And Main Contributions:**

The central aim of this paper is to address data bias by introducing Bias Experts, which are a set of binary classifiers. The authors conducted a preliminary study to validate that the predictions made by the Bias Experts can more accurately indicate whether an example is biased or not. By simultaneously training the Bias Experts with the main model using PoE, the proposed method achieves improved model performance on biased test sets. The experimental results substantiate the effectiveness of this approach. However, further verification of the motivation is warranted, and some recent comparison methods are missing.

------ Post Rebuttal ------
The authors' response resolved my concerns, prompting me to raise the soundness score from 3 to 4.

**Questions For The Authors:**

a. While the preliminary study confirms that bias experts can effectively reflect the degree of bias in samples, the relationship between these bias experts and a better-debiased model remains underexplored. For instance, does a "more biased" auxiliary model lead to a more robust main model? From my perspective, a more biased auxiliary model might introduce bias in the opposite direction to the main model. Consequently, the debiased model may inaccurately predict biased test samples due to potential excessive penalization, adversely affecting performance on in-domain test sets.

b. Some recent comparison methods published after 2021, such as Ghaddar et al., 2021, and Lyu et al., 2023, are absent in the current study.

c. The preliminary study includes experiments on biased/bias-conflicting examples. However, I couldn't find any description regarding how these examples were obtained. It would be helpful to include information about the methodology used to gather these examples for the experiments.

[1] Ghaddar, A., Langlais, P., Rezagholizadeh, M., & Rashid, A. (2021). End-to-end self-debiasing framework for robust NLU training. ACL/IJCNLP (Findings) 2021

[2] Lyu, Yougang, et al. "Feature-level debiased natural language understanding." AAAI 2023

**Reasons To Accept:**

a. The introduction of a set of binary classifiers is novel in the field of debiasing.

b. The experimental results provide evidence of the proposed method's superiority when compared with the listed baselines.

**Reasons To Reject:**

a. The motivation should be further verified. (See question a)

b. Some recent comparison methods are missing. (See question b)

**Reproducibility:**

4: Could mostly reproduce the results, but there may be some variation because of sample variance or minor variations in their interpretation of the protocol or method.

**Reviewer Confidence:**

4: Quite sure. I tried to check the important points carefully. It's unlikely, though conceivable, that I missed something that should affect my ratings.

---

> ### Author Rebuttal · Authors · 2023-08-29
>
> **Question 1.** The motivation should be further verified. The relationship between these bias experts and a better-debiased model remains underexplored. For instance, a more biased auxiliary model might introduce bias in the opposite direction to the main model. Consequently, the debiased model may inaccurately predict biased test samples, adversely affecting performance on in-domain test sets.
>
> **Answer.** Thanks for the constructive comment. The motivation of this paper is the development of a more effective "training methodology" for the biased model to improve the trade-off in the debiased model between OOD performance and ID performance. Developing better training methods can mitigate the ID performance degradation you're concerned about. Specifically, as shown in the attached table below, our model can achieve the same OOD performance gain at the expense of less ID performance degradation. Degradation of ID performance can be controlled in various ways, by reducing the weight of PoE loss in a multi-loss [1], or by training the biased model with fewer steps.
>
> |                          | ID (MNLI) | OOD (HANS) |
> |------------------------|:---------:|:----------:|
> | PoE (Utama et al., 2020) |    81.9   |    66.8    |
> | PoE (Sanh et al., 2021)  |    83.3   |    67.9    |
> | PoE + Ours (2k)          |  **83.8** |    68.0    |
> | PoE + Ours (Full)        |    82.7   |  **72.6**  |
>
> Here, "2k" in the 3rd row means training the main model for 2k steps, and "Full" means in the 4th row means training the main model for 3 epochs, as detailed in the section 4.3.
>
> [1] Sanh et al., Learning from others' mistakes: Avoiding dataset biases without modeling them, ICLR 2021.
>
> &nbsp;
>
> **Question 2.** Some baselines are missing, such as Ghaddar et al., 2021, and Lyu et al., 2023.
>
> **Answer.** Ghaddar et al. (2021) has been compared with our model in Table 1 of the submitted version, and is denoted by Self-Debiasing [2]. Meanwhile, we excluded DCT [3], due to the scope difference. While we focus on improving the learning of the biased model, DCT focuses on improving the utilization of the existing biased model.
>
> [2] Ghaddar et al., End-to-end self-debiasing framework for robust NLU training, Findings of ACL-IJCNLP 2021.\
> [3] Lyu, et al., Feature-level debiased natural language understanding, AAAI 2023.
>
> &nbsp;
>
> **Question 3.** Details about the methodology for classifying biased/bias-conflicting examples in the preliminary study would be helpful.
>
> **Answer.** Thank you for the comment. We have collected biased/bias-conflicting examples from the training set of MNLI and FEVER. For MNLI, we have filtered examples where all the hypothesis words occur in the premise by exact match, then have classified the entailment class as biased and others as bias-conflicting. For FEVER, we have filtered examples where any of the Top 10 LMI-ranked bigrams (listed in Schuster et al. (2019) [4]) appear in the claim (also by exact match), then have classified the refute class as biased and others as bias-conflicting. We will attach details about the methodology for classifying examples in the camera-ready version.
>
> Additionally, we attach a Python code named “train_bert_poe_track.py” to Github, which has been used in our preliminary study. Please see lines 713-733 of this code for the detailed implementation of classifying biased/bias-conflicting examples.
>
> [4] Schuster et al., Towards Debiasing Fact Verification Models, EMNLP 2019.
>
> &nbsp;
>
> **Missing Reference.** We will update the related work section with the missing references you pointed out.

---

### Official Review · Reviewer_JToF · 2023-08-09

**Soundness:** 4

**Excitement:**

4: Strong: This paper deepens the understanding of some phenomenon or lowers the barriers to an existing research direction.

**Paper Topic And Main Contributions:**

This document discusses a debiasing framework for natural language understanding (NLU) models. It addresses the issue of biases in datasets that can lead to high performance on in-distribution data but poor performance on out-of-distribution data. The proposed framework introduces bias experts and binary classifiers trained between the main and auxiliary models. These bias experts bridge the gap between the auxiliary model's learning objective and purpose, resulting in improved performance on various challenge datasets.
Key points are:

1. Biases in datasets can cause NLU models to perform well on certain data distributions but poorly on others.
2. Previous debiasing methods down-weight biased examples identified by an auxiliary model trained with explicit bias labels.
3. The proposed framework introduces bias experts, and binary classifiers trained between the main and auxiliary models.
4. Each bias expert is trained on a binary classification task derived from the multi-class classification task using the One-vs-Rest approach.
5. Experimental results show that the proposed strategy effectively reduces the gap between in-distribution and out-of-distribution performance and improves performance on various challenge datasets.


**Questions For The Authors:**

While the code for the experiments is easily accessible, I wonder whether there is an easy way of integrating your idea into existing systems. Right now, the ease of access outside of your experiment is limited.

**Reasons To Accept:**

1. The paper focuses on a critical and timely issue in Natural Language Understanding (NLU) - debiasing models. The topic is of significant interest given the increasing awareness of biases in machine learning models and their potential societal implications.

2. The paper introduces a method of learning debiased classifiers with the help of biased committees. This approach seems to be a novel contribution to the field, offering a different perspective on tackling biases.

3. The paper evaluates the proposed model and baselines on three NLU tasks with out-of-distribution evaluation sets: Natural Language Inference (NLI), fact verification, and paraphrase identification. Such a comprehensive evaluation provides a robust understanding of the model's performance across different tasks.

The paper offers substantial reproducibility, is thoroughly edited, and contains a valid and important ethical statement.


**Reasons To Reject:**

The paper acknowledges that their method's high performance on out-of-distribution data (new or unseen data) comes at the expense of in-distribution performance (data it was trained on). This trade-off could be a concern, especially if the in-distribution performance drop is significant. I also see the memory usage limitation, but without knowing it in detail, I would guess that applications exist where this approach is still usable and helpful.



**Reproducibility:**

5: Could easily reproduce the results.

**Reviewer Confidence:**

2: Willing to defend my evaluation, but it is fairly likely that I missed some details, didn't understand some central points, or can't be sure about the novelty of the work.

**Typos Grammar Style And Presentation Improvements:**

Line 315: "Natural language inference (NLI) is the task of determining whether a premise entails, contradicts, or is neutral with a hypothesis given a sentence pair." - The phrase "neutral with a hypothesis" might be clearer as "neutral to a hypothesis".
Line 324: "Fact verification is the task of determining whether the information in the evidence supports a claim, refutes a claim, or does not contain enough information." - Consider rephrasing for clarity: "Fact verification involves determining if the evidence supports or refutes a claim, or if it lacks sufficient information."
Line 303: The "Experiments" section could benefit from subheadings for each task (NLI, Fact Verification, Paraphrase Identification) to make it clearer and easier to follow.
Line 335: The description of the "Paraphrase identification" task ends abruptly, and the reader is left without a clear understanding of the evaluation metrics or datasets used. This section should be completed for clarity.
Overall: I feel PoE could be detailed with another sentence or two. Not everyone is familiar with it.

---

> ### Author Rebuttal · Authors · 2023-08-29
>
> **Reason to reject 1.** The trade-off between out-of-distribution and in-distribution performances could be a concern.
>
> **Answer.** We agree that the trade-off could be a concern in debiasing works, as in previous works. However, our method achieves the best trade-off between out-of-distribution and in-distribution performances. The results are shown in the table below. Specifically, the proposed method shows the largest OOD (HANS) performance improvement at the expense of the same or less ID (MNLI) performance.
> |                          | ID (MNLI) | OOD (HANS) |
> |------------------------|:---------:|:----------:|
> | BERT-base                |    84.5   |    62.4    |
> | PoE (Utama et al., 2020) |    81.9   |    66.8    |
> | PoE (Sanh et al., 2021)  |  **83.3** |    67.9    |
> | PoE + Ours (10k)         |  **83.3** |  **71.0**  |
>
> Here, "10k" in the 4th row means training the main model for 10k steps.
>
> &nbsp;
>
> **Reason to reject 2.** I also see the memory usage limitation, but without knowing it in detail, I would guess that applications exist where this approach is still usable and helpful.
>
> **Answer.** Yes, there could be a memory usage limitation (lines 219-225) since we need as many bias experts as the number of classes of the task. However, most NLU tasks, such as fact verification and natural language inference, have a moderate number of classes (2~20, See footnote 2), and the bias experts are used only in training. We demonstrate the applicability of our method to various NLU tasks in the paper.
>
> &nbsp;
>
> **Question 1.** While the code is easily accessible, the ease of access outside of your experiment is limited.
>
> **Answer.** Thanks for the comment. We will revise the code to easily integrate the proposed method into existing systems.

---

### Meta-Review · Area_Chair_unbt · 2023-09-20

**Recommendation:** 4

**Metareview:**

This paper presents a debiasing framework without prior knowledge of the bias. This is a challenging but realistic setting. In particular, they proposed to use one-against-all (i.e., multiple binary classifiers) to construct a bias-only model that better captures the bias in the dataset.

Pros:
- The paper focuses on a critical, challenging, and realistic problem. Specifically, several prior work in bias mitigation assumes type of biases is known but this paper does not make such an assumption.
- The proposed approach using a set of binary classifiers to capture biases is reasonably novel.
- Experiments are sufficient to support the claim.

Cons:
- It is unclear whether the approach is practical given the tradeoff between in-distribution and out-distribution performance.
- Experiments on the BERT-based model are a bit out-of-date. Although it's understandable that the authors want to compare with prior work, it would be interesting to show performance on more recent LLMs.

---

### Decision · Program_Chairs · 2023-10-07

**Decision:**

Accept-Main

**Comment:**

This paper presents a debiasing framework without prior knowledge of the bias. This is a challenging but realistic setting. In particular, they proposed to use one-against-all (i.e., multiple binary classifiers) to construct a bias-only model that better captures the bias in the dataset.

Pros:
- The paper focuses on a critical, challenging, and realistic problem. Specifically, several prior work in bias mitigation assumes type of biases is known but this paper does not make such an assumption.
- The proposed approach using a set of binary classifiers to capture biases is reasonably novel.
- Experiments are sufficient to support the claim.

Cons:
- It is unclear whether the approach is practical given the tradeoff between in-distribution and out-distribution performance.
- Experiments on the BERT-based model are a bit out-of-date. Although it's understandable that the authors want to compare with prior work, it would be interesting to show performance on more recent LLMs.